# Your head is there to move you around: Goal-driven models of the primate dorsal pathway

**Patrick J Mineault**
patrick.mineault@gmail.com

**Shahab Bakhtiari**
Mila, McGill University
bakhtias@mila.quebec

**Blake A Richards**
Mila, Montreal Neurological Institute, McGill University
blake.richards@mila.quebec

**Christopher C Pack**
Montreal Neurological Institute, McGill University
christopher.pack@mcgill.ca

## Abstract

Neurons in the dorsal visual pathway of the mammalian brain are selective for motion stimuli, with the complexity of stimulus representations increasing along the hierarchy. This progression is similar to that of the ventral visual pathway, which is well characterized by artificial neural networks (ANNs) optimized for object recognition. In contrast, there are no image-computable models of the dorsal stream with comparable explanatory power. We hypothesized that the properties of dorsal stream neurons could be explained by a simple learning objective: the need for an organism to orient itself during self-motion. To test this hypothesis, we trained a 3D ResNet to predict an agent's self-motion parameters from visual stimuli in a simulated environment. We found that the responses in this network accounted well for the selectivity of neurons in a large database of single-neuron recordings from the dorsal visual stream of non-human primates. In contrast, ANNs trained on an action recognition dataset through supervised or self-supervised learning could not explain responses in the dorsal stream, despite also being trained on naturalistic videos with moving objects. These results demonstrate that an ecologically relevant cost function can account for dorsal stream properties in the primate brain.

## 1 Introduction

The mammalian visual cortex is organized into two processing streams [1]: the ventral stream, where neurons are selective for object class and identity; and the dorsal stream, where neurons are selective for motion. Neurons in the ventral stream exhibit selectivity for increasingly complex stimulus features at successive stages, from oriented lines in V1, to textures in V2, curved lines in V4, and culminating in representations of natural objects in the inferotemporal (IT) cortex [2, 3, 4, 5]. The myriad response properties within and across stages have been difficult to understand computationally [6].

However, in recent years, a large body of work [7, 8, 9, 10, 11, 12, 13] has found that modern convolutional neural networks trained on image classification develop representations that match those found in the ventral stream. Early CNN layers match primary visual cortex (V1), while higher-level layers better match higher-level ventral stream areas, both in terms of qualitative preferred

35th Conference on Neural Information Processing Systems (NeurIPS 2021).

features [12] and quantitative predictions of responses to arbitrary stimuli [13, 14]. Moreover, models which perform well on ImageNet image classification tend to explain a larger proportion of the variance in ventral stream area IT [8]. It has recently been found that high-performing networks can emerge through more biologically plausible self-supervised training [15, 16]. These results make it possible to interpret the sometimes baffling data about neural responses in the ventral stream in terms of a biologically plausible distributed learning algorithm whose goal is to develop invariant representations that can support object recognition behavior [9, 17].

Although this approach has been similarly fruitful in other domains (e.g., audition [18, 19]), it has not yet been applied to the dorsal visual pathway. From physiological recordings in dorsal stream areas like MT and MST [20, 21, 22], we know that neurons in this pathway are exquisitely selective for motion and increase in receptive field size and complexity along their hierarchy. These properties have inspired different conceptions of dorsal pathway function, including action recognition [23, 24], prediction of image sequences [25], and tracking of object motion [26], to name just a few. At present, there is no way to know which, if any, of these proposals is correct.

We hypothesized that dorsal pathway representations emerge from a simple objective: the need for the organism to orient itself during self-motion. As animals move through the world, they must estimate the parameters of their own motion, in order to avoid collisions, to plan trajectories, and to stabilize their gaze on objects of interest; the latter is critical for maintaining visual acuity. We suggest that this can be accomplished by learning, in a self-supervised way, the relationship between retinal images and self-motion parameters inferred from oculomotor and vestibular signals that exist in the brain [27] [28]. To test this hypothesis, we trained a 3D ResNet to predict the parameters of simulated self-motion - walking speed and head rotation – in short sequences of motion through simulated environments. We found that this network, dubbed *DorsalNet*, learned motion representations that were qualitatively similar to those found in the dorsal visual stream. Specifically, units were tuned for local motion direction in the earliest layers, object motion in intermediate layers, and complex optic flow in the highest layers [29].

To test our hypothesis quantitatively, we built a database of neural recordings from different regions of the dorsal visual pathway in non-human primates [14]. We then compared the ability of different networks to explain responses in areas V1, MT, and MST. We found that DorsalNet consistently outperformed 3D ResNets trained on action recognition in a supervised manner. Both the self-motion estimation objective and the training stimulus seemed to be critical, since 3D ResNets trained with a predictive objective [CPC; 30] or supervised on action sequences showed weaker performance. Thus, we demonstrate that the diverse neural response properties in the dorsal stream can be captured by a network that has the goal of estimating self-motion parameters from natural image sequences, both elucidating the functional role of the dorsal stream and creating a best-in-class, in-silico model of the dorsal stream.

## 2   Background and related work

**Dorsal stream processing**   The dorsal stream - also known as the *where* pathway - is a network of cortical areas that are selective for visual motion (Figure 1A). It originates in primary visual cortex (area V1) with a subpopulation of neurons that respond selectively to oriented edges moving in a particular direction. These cells project to areas MT/V5 [27, 31], where most neurons respond selectively to motion direction, even for relatively complex stimuli comprised of multiple edges or features [21, 32]. These neurons in turn project to area MST, where many neurons are selective for the kinds of complex motion patterns that arise during locomotion [22]. MST is considered the terminal stage of the dorsal stream, with subsequent areas integrating information from other senses to support diverse roles in action recognition [23], decision-making [33], and spatial memory [34].

**Models of the dorsal stream**   Previous models of the dorsal stream have emphasized different possible functions. Giese and Poggio [23] have argued that the progression of selectivity along the pathway is well-suited to the recognition of biological movements, and this is consistent with studies showing that the ability to identify shapes from motion patterns is disrupted by lesions to area MT [35]. Other models have posited a role for the dorsal pathway in segmenting moving objects [36, 37], predicting future image frames [25], or supporting reaching movements [38]. Finally, a body of computational [39, 40] and experimental [41] work has analyzed the potential role of dorsal stream

neurons in the perception of heading or path [42]. None of these models has been quantitatively compared to the detailed properties of neural responses in the dorsal stream.

Other models have made this kind of comparison, but they have been based on shallow architectures and fit directly to the data from dorsal stream areas, including V1 [43], MT [44, 45, 46] and MST [47]. Although these models shed light on the mechanisms by which neurons attain their stimulus selectivity, they do not relate in any clear way to the functional hypotheses mentioned above.

We have therefore attempted to link the properties of dorsal stream neurons, obtained from a database of recordings in non-human primate cortex, to specific functional objectives hypothesized in previous work. In this sense our work is in line with the goals of BrainScore [14], which seeks to benchmark ANNs by their ability to explain ventral stream neurons, and to recent work examining self-supervised networks' fits to ventral areas [16, 15].

## 3    Methods

**Training network for self-motion**    We generated a dataset consisting of short videos (10 frames) of self-motion in AirSim, a package for drone and land vehicle simulations in Unreal Engine [48]. These videos simulated walking along linear trajectories with constant head rotations in two environments (Figure 1B), starting at random positions, varying environmental conditions, hour of day, starting head pose speed and walking speed (Table S1 in the Appendix). Sequences that led to collisions with the environment were removed.

We trained a 6-layer 3D ResNet (layer definitions in table S1) to predict 2 of the components of head rotation (yaw and pitch rotation speed; roll was not simulated) and the 3 components of linear velocity (parametrized as yaw and pitch heading and speed). We chose a 3D ResNet architecture over alternatives for its stable training, wide use in video tasks [49, 50], and the high performance of 2D resnets in modeling the ventral stream [14]. We discretized each component into 72 bins and trained the network with a cross-entropy objective for each of the 5 components. We used the Adam optimizer with a step size of 0.003, batch norm, and trained for 100 epochs.

**Neural datasets**    Datasets are listed in Table 1. All experiments were conducted in non-human primates (macaca fascicularis and macaca mulatta) and were approved by the governing IRB; detailed experimental procedures are available in the corresponding publications. Data are used under the license terms listed on crcns.org or by permission from the authors [47]. Methods varied from dataset to dataset, but generally, non-human primates were instructed to fixate on a small target while a contiguous image sequence was presented for several minutes. In some cases, parts of the image sequence were repeated when fixation was lost. Image sequences consisted of color movies, black-and-white movies, static pictures with simulated motions, and random dot kinematograms. Data was collected using single electrodes or multi-electrode arrays. Where available, we used previously published sorted spikes; when spikes were unsorted, we used multi-unit activity.

We split each dataset into a train and test set [51]; when only a subset of these stimuli were repeated several times, or a dataset had a designated test subset, we used this subset as the test set; in other cases, we split the data into 6-second blocks, and concatenated every 10th block to form a test set. We kept the sampling rate of the image sequence at its natural rate and resampled neural activity at the same rate, indicated in Table 1. We resampled all stimuli spatially to 112x112. In control analyses, we resampled the input to 74x74 or 168x168 to measure the sensitivity of the results to scale. In the case of [47, 44], the seeds originally used to determine the exact location of dots in the random dot kinematograms were lost, hence we regenerated stimuli with dots in different locations; it should be noted that this could limit the maximal performance of networks [52]. All data used in this paper has been previously published; we release preprocessing scripts and PyTorch loaders to facilitate replication.

**Aligning ANNs and neural activity**    We computed latent representations at different layers of the target ANNs, listed in Table 2, in windows of 10 image frames preceding neural activity. We cropped the first and last latent activity frame and downsampled the activity 2-fold temporally to obtain 4 frames of latent representations preceding the neural activity, and spatially averaged and downsampled each layer output to 8 by 8. Following [14], we kept the first 500 PCs of the intermediate representation and used ridge regression to find mappings from latent space to experimental neural

activity. We selected the ridge parameter using 5-fold cross-validation within the train set. Where test sets with 5 or more disaggregated repeats were available, we report an R score normalized against the maximum attainable R score [53]; otherwise, we report the raw R score.

In control analyses, we replaced ridge regression with a sparse regression estimated through boosting. To fit all the intermediate representations in memory and fit a boosted regression model, we used two different strategies to reduce the memory footprint: downsampling layer outputs (with spatial averaging as in the linear regression; table S4 in the appendix) or subsampling (without spatial averaging; Table S4). We selected the number of boosting iterations using 5-fold cross-validation within the train set. We fit these models on a commodity GPUs including P5000 and 1080Ti locally, in Paperspace and in AWS for a total of $\sim 1000$ single-GPU-hours. Model weights and code are available [1] under an MIT license.

**Contrastive Predictive Coding (CPC)**    CPC is a self-supervised learning algorithm that learns to predict the next latent state of a sequence (e.g. a video sequence) given its present and past states. The details of the CPC algorithm can be found in [30] and [54], but we summarize it briefly here. A sequence of video frames $(x_t)$ are passed as input to a 3D CNN. The CNN output $(z_t)$, which is a latent representation of the video sequence, is fed to a recurrent neural net (RNN). The RNN aggregates past and present latent states (i.e. CNN output) and generates a context variable as its output $(c_t)$. The context variable is then passed to a single layer MLP which predicts the future latent state of the video. The predicted latent state and the true latent state (positive pairs), along with some incorrect examples of the next state (negative pairs) are given to a contrastive loss function. Minimizing the contrastive loss maximizes the similarity of the predicted and the true next states, and minimizes the similarity of the predicted and the false next states.

## 4    Results

### 4.1    3D resnets trained for self-motion learn dorsal-like representations

We hypothesized that learning to estimate self-motion from visual inputs would lead to dorsal stream-like representations. As in the ventral stream, these representations begin in V1 with receptive fields that encode simple, local features of stimuli. Through subsequent recombinations at different layers, more complex and ecologically relevant encoding emerges. To test this hypothesis, we generated self-motion videos in a simulation environment, and trained a 3D ResNet to predict its self-motion parameters, namely head rotation and linear locomotion (see Methods for details).

**Qualitative matches to the dorsal stream**    The 6-layer 3D ResNet trained in this way learned representations similar to single units in the primate dorsal stream. We focus our attention here on layers 1, 2 and 3 of the network. Preferred features of layer 1 contained many spatiotemporally slanted filters (Figure 1C), which are the building blocks of motion selectivity in primate V1 [55]. We quantified this slant with the separability index $\sigma_1^2 / \sum_i \sigma_i^2$ from the singular values of the grayscale filters $\sigma_i$; this matched values reported in the literature for V1 [20] [.72 +/-.16 for trained network, .71 +/- .15 in real V1 neurons; figure S1 in the appendix].

To gain insight into the stimulus selectivity of these representations, we generated optimal stimuli for individual units in intermediate layers of the network by optimization [12]; we present static images of the intermediate preferred frame here, while animations can be visualized on the companion website[2]. Probed in this fashion, many intermediate features in layer 1 preferred what looked like drifting gratings (examples in Figure 1D), consistent with the selectivity of V1 cells [55, 56]. Hence, to further probe the selectivity of these units, we used full contrast, drifting gratings of different spatial and temporal frequencies, placed in the center of the visual field. Tuning curves in layer 1 (samples in Figure 1) tended to have a bias towards direction selectivity, with a mean circular variance at the preferred spatial and temporal frequency of 0.75 and a median direction selectivity index - defined as $1 - r_{pref}/r_{antipref}$ on the centered tuning curves - of 0.98. This is somewhat higher than is typically found in V1 [57], likely due to lack of noise, but it is close to the selectivity of the V1 neurons that actually project to higher levels of the dorsal visual pathway [58].

---

[1]`https://github.com/patrickmineault/your-head-is-there-to-move-you-around`
[2]`https://your-head-is-there-to-move-you-around.netlify.app`

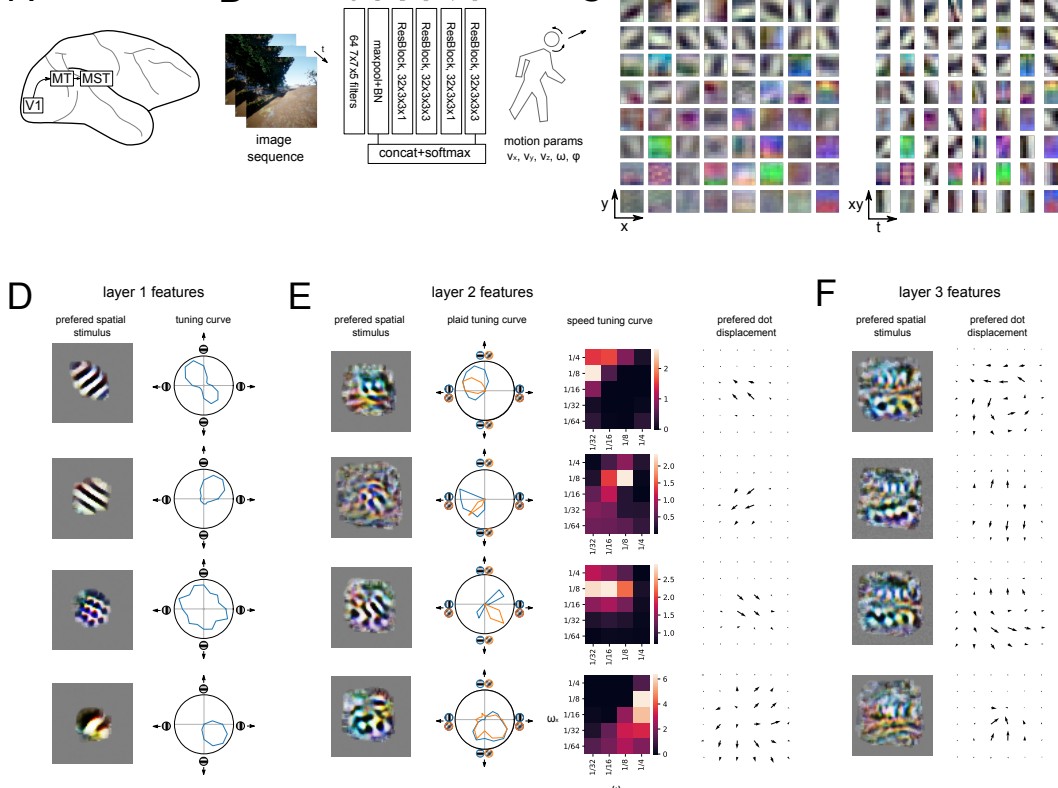

Figure 1: A 3D ResNet trained for self-motion estimation learns dorsal-like representations. A. The goal of this study is to model the dorsal visual stream, including V1, MT and MST. B. The 3D Resnet model is trained to estimate self-motion parameters from image sequences. C. Weights of the first layer. First layer filters are selective in space-time. D. Sample tuning of layer 1 features. Layer 1 contains many direction-selective cells reminiscent of V1. E. Sample tuning of layer 2 features. Many layer 2 features exhibit tuning for rigid motion, similar to MT. F. Sample tuning curve of layer 3 features. Many cells in layer 3 are tuned for complex optic flow, like MST

Layer 2 units tended to prefer more spatially broadband moving stimuli, not unlike the plaids conventionally used in probing MT cells [21] (Figure 1E, left column). Indeed, probing the representations with sums of gratings revealed similar selectivity to a single grating in a subset of cells (Figure 1E, middle column; pattern selectivity plots in Figure S1 in the appendix). These cells likely encode stimulus velocity in a manner that is invariant of the composition of the stimulus [21]. Like MT cells, subunits in this layer tended to be highly direction selective, with the average circular variance of the direction tuning curves being .41.

MT cells are also known to be selective for stimulus speed, which is the ratio of temporal to spatial frequencies [59]. A similar kind of selectivity emerged in layer 2 of the model, where many units preferred higher temporal frequencies when the spatial frequencies were higher (example tuning curve in Figure 1E). To quantify this selectivity, we probed the model units with a range of spatiotemporal frequencies and fit the data with slanted Gaussian functions [60], which revealed a mean speed selectivity index of -.14 in layer 1, compared to 0.58 in layer 2, the latter being similar to the value of 0.52 reported in MT [60]. Probing layer 2 units with moving dots, we found a majority of neurons with simple receptive fields that prefer linear motion, with a smaller number of complex receptive fields (Figure 1E, bottom right).

Finally, we found many cells in layer 3 that combined the outputs of lower-level units to generate selectivity for more complex motion patterns (example cells in Figure 1F). Dot pattern probes revealed selectivity for rotations, spirals or single axis expansion. As in primate area MST, these units tended

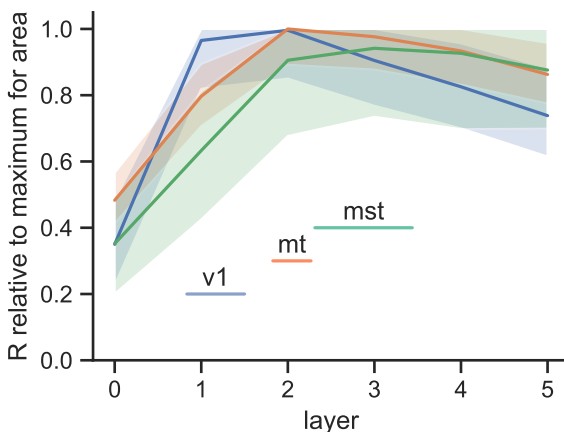

Figure 2: Layers 1, 2, and 3 of DorsalNet best match areas V1, MT and MST. Lines show correlation (R) relative to maximum for area. Horizontal lines: 95% CI of layer with maximal alignment to area.

to emphasize expansion motion rather than contraction, similar to the bias experienced during forward navigation (Figure S1) [61].

| Area | Dataset | Data | Sampling | Stimulus |
|------|---------|------|----------|----------|
| V1 | crcns-pvc1 [62, 63] | 23 multi-units | 30Hz | Color movies |
| | crcns-pvc4 [64, 65, 66] | 25 single units | 75Hz | B&W movies |
| MT | crcns-mt1 [44, 67] | 88 single units | 30Hz | optic flow kinematograms |
| | crcns-mt2 [46, 68] | 44 single units | 83Hz | B&W motion-enhanced movies |
| MST | packlab-mst [47] | 36 single units | 30Hz | optic flow kinematograms |

Table 1: Datasets

**Regression analysis of representations**    Given that the trained network recapitulated many qualitative properties of the dorsal stream, we next investigated whether they quantitatively matched dorsal stream areas, for which single-neuron data was available. We used ridge regression to learn a mapping from latent representations at each layer of the network to single neural responses to complex stimuli, including black and white and color movies, along with random dot kinematograms (See Methods and Table 1 for details). We learned a separate mapping for each layer of the network, allowing us to match the depth of the network to each brain area. As seen in Figure 2, this showed a hierarchical progression, with higher-level cortex matching higher-level layers in the ResNet. The average best matching layer across cells with report correlation greater than .01, illustrated by the horizontal lines, was 1.1 for V1 cells [(0.9, 1.5) 95% CI, bootstrap across cells], 2.0 for MT cells [(1.8, 2.3)] and 2.9 for MST cells [(2.3, 3.4)].

We noticed that the mapping was less distinct for layer 3. We investigated this further by measuring the response of the network to a gauntlet of stimuli from the Airsim dataset. Centered kernel alignment (CKA) [69] revealed that while layers 0, 1, and 2 had highly distinct representations, subsequent layers were less distinct S3. We also investigated the robustness of the mapping to a change of scale of the input sequences [70]. We saw some minor shifts: the median mean layer assignment for V1 was 1.0 at 0.66X scale, 1.1 at the standard 1X scale, and 1.3 at 1.5X scale (Figure S2). Overall, however, mappings were robust to a change of scale. Thus, broadly speaking, layers 1, 2 and 3 of the network recapitulated V1, MT and MST, respectively.

## 4.2   Networks with alternative objectives do not account for responses in the dorsal stream

**Action recognition networks**    To examine the specificity of these results, we tested other networks trained with different objective functions. Action recognition is a popular computer vision task,

| Category | Name | Dataset | License | Notes |
|---|---|---|---|---|
| SlowFast [49] | slowfast
i3d | Kinetics400 | Apache | Fast branch only |
| R3D [50] | r3d_18
r2plus1_18
mc3_18 | Kinetics400 | BSD | |
| CPC [30] | cpc_ucf
cpc_airsim | UCF101
Airsim | own work | R3D with 10 res blocks
R3D with 10 res blocks |
| Gabors [46] | gabor
gabor_nomotion | -
- | own work | Opposite dirs averaged |
| MotionNet [26] | motionnet | shifted images | CCBY4.0 | |
| DorsalNet | dorsalnet | Airsim | own work | R3D with 4 res blocks |

Table 2: Models tested

| | V1 | | MT | | MST |
|---|---|---|---|---|---|
| | pvc1 | pvc4 | mt1 | mt2 | mst |
| slowfast | **.471** (.034) | .361 (.042) | .211 (.018) | .281 (.015) | .189 (.044) |
| i3d | **.457** (.036) | **.389** (.046) | .213 (.018) | .284 (.015) | .219 (.044) |
| r3d_18 | .403 (.032) | **.383** (.042) | .217 (.018) | .289 (.015) | .224 (.046) |
| r2plus1d_18 | .428 (.035) | **.382** (.042) | .215 (.018) | .282 (.015) | .226 (.043) |
| mc3_18 | .405 (.034) | **.393** (.045) | .218 (.018) | .276 (.014) | .228 (.045) |
| cpc_ucf | .271 (.044) | **.394** (.046) | .214 (.018) | .241 (.016) | .190 (.045) |
| cpc_airsim | .422 (.036) | **.384** (.045) | **.250** (.020) | .360 (.017) | .292 (.045) |
| gabor_nomotion | .273 (.035) | .353 (.038) | .212 (.018) | .188 (.014) | .248 (.045) |
| gabor | .325 (.036) | **.366** (.037) | **.249** (.019) | .301 (.015) | .394 (.054) |
| motionnet | .276 (.042) | **.364** (.039) | .238 (.018) | .333 (.016) | **.441** (.053) |
| dorsalnet | .364 (.043) | **.370** (.039) | **.251** (.019) | **.381** (.017) | **.454** (.054) |

Table 3: DorsalNet quantitatively performs best across the dorsal stream. Table shows normalized pearson correlation (R; see Methods for definition) of different models on different datasets. In parenthesis: standard error of the mean over cells.

and so we tested 3D ResNets trained on Kinetics400 [71]. These networks performed admirably in explaining V1 responses, reaching an average R > .4 on the pvc1 dataset. However, across our MT and MST datasets, performance was poor, failing to exceed that of a null model [72] consisting of a 3D Gabor pyramid (Table 3). We note that only a small fraction of V1 neurons project to the dorsal stream, with the majority projecting to ventral stream areas; we interpret the relative performance in V1 vs. MT and MST as a sign that these networks learned representations more aligned with the *ventral* stream, supporting object recognition and by extension action recognition. Consistent with this interpretation, we found that the first layer of 3D ResNets trained for action recognition did not learn motion in the traditional sense (Figure S5). Instead, their filters were mostly separable in space and time, meaning they were not selective for motion energy *per se*.

**CPC**   Our results indicate that learning to estimate self-motion in a simulated environment creates representations similar to those in the primate dorsal stream. The neural network architecture (3D ResNets) was similar for the self-motion estimation objective and action recognition tasks. However, both the task - prediction of self-motion parameters - and the stimulus ensemble - self-motion sequences in the Airsim environment - differed. To tease apart the relative importance of these two factors, we tested the ability of contrastive predictive coding (CPC) networks [30] to account for responses in the dorsal stream when trained over different stimulus ensembles. CPC is a self-supervised training method that finds predictive latent representations that can distinguish

between image sequences. Importantly, it is possible to apply the CPC objective to different stimulus ensembles, thereby differentiating between task and stimulus ensemble effects. We trained an 11-layer network with a CPC objective on the UCF101 dataset and our Airsim dataset. The Airsim-trained network performed significantly better than the UCF101-trained network, approaching the performance of DorsalNet in MT but not in MST. Examining first layer filters revealed direction-selective receptive fields after training on the Airsim dataset but not with UCF101 (Figure S5). This is consistent with the training set being necessary, though not sufficient, to match primate dorsal stream neurons.

**MotionNet**    We next tested a much simpler 2-layer network from the neuroscience literature, which was trained to estimate the linear motion of black and white image patches [26]. The original model was a fully connected architecture working on small image patches, and we made it convolutional by tiling. We used the checkpoints shared by the authors as the model weights. This model had not previously been directly benchmarked against neural data, and given the small size of its stimulus ensemble, we did not expect it to perform well. Surprisingly, it scored far better in predicting MT and MST responses than action recognition networks (Table 3). We found in a control analysis (Table S3 in the appendix) that the relative performance of MotionNet could be improved still by spatially scaling up the stimulus, matching the performance of DorsalNet on 2 out of 3 MT and MST datasets. These results are consistent with solving 2D motion being an important sub-goal of the dorsal stream.

We next asked whether there existed a one-to-one or few-to-one relationship between model subunits and single neurons. Using sparse regression, we found that DorsalNet better matched individual neurons across all MT and MST datasets than MotionNet, regardless of scaling (Tables S4 and S5 in the appendix). Thus, DorsalNet subunits were more directly aligned to single neurons across the dorsal stream.

## 5    Self-motion estimation performance correlates with dorsal stream match

Across our baselines, there was a large range in the ability of different models to reproduce dorsal stream data. We asked whether this heterogeneity could be linked to performance on a self-motion estimation task. We froze the weights of our baseline networks and trained linear decoders to estimate self-motion parameters on the AirSim dataset from hidden layer representations. We excluded DorsalNet and Airsim-trained CPC from these comparisons. Across our baselines, there was a highly significant correlation between self-motion estimation performance and match to MT and MST neurons (Figure S4; Table S2 in the appendix). Interestingly, when looking at individual self-motion parameters, head rotation estimation accuracy was most correlated with performance on MT and MST datasets. Thus, those networks which happen to be best at self-motion estimation, especially head rotation, can best explain responses in the dorsal stream, consistent with a formative role of self-motion estimation in dorsal stream representations.

## 6    Limitations

**Multiple interpretations**    We show that learning to estimate one's self-motion from visual cues leads to representations which are similar to those of the dorsal stream. We benchmark against several other candidate models, including localized frequency detectors, which form a sparse basis for images [73], predictive coding models, models trained for action recognition, and models trained trained to estimate the motion of small image patches. While DorsalNet performed best overall across the dorsal stream, we found that MotionNet and a CPC-based network trained on our AirSim dataset were close contenders. With the available data, we cannot conclusively rule out that these alternative objectives, with the right tweaks, could not account for the data. One interesting possibility is that, as MotionNet hints, solving rigid 2D motion is a sub-goal of the dorsal stream; and, as DorsalNet shows, the supervisory signal needed to learn to solve that sub-goal could come from head movements, especially head rotations, via efference copy. An open benchmark in the style of [14] could reveal other objectives compatible with the data and refine these results.

**Data limitations**    To the best of our knowledge, we used all of the relevant publicly available non-human primate data for this study. Most of this data was collected more than a decade ago in time-consuming single-electrode experiments, with electrode drift, loss of fixation, short recording

times and small numbers of recordings per experiment being significant limitations. The MST dataset in particular is not very discriminative across models. Differences in stimuli and number of repetitions make absolute comparisons across areas difficult. Improvements in recording technology as well as better-designed hypothesis-driven studies will allow the collection of more discriminative data in the future. Our study paves the way for closed-loop experiments to verify that the estimated stimuli indeed maximally drive dorsal stream neurons [74, 75].

# 7 Discussion

Systems neuroscience aims to explain how the brain solves behavioral tasks at the algorithmic level [17]. While a rich literature has linked the ventral visual stream to the task of object recognition, little work has focused on understanding how and why dorsal streams representations emerge. Noting the critical role of self-motion estimation across the animal kingdom [76], we hypothesized that training an artificial neural net on self-motion estimation from image sequences would lead to representations similar to the dorsal stream. We verified this qualitatively by probing networks with artificial stimuli and by finding maximizing stimuli. We confirmed these findings quantitatively by benchmarking existing computer vision networks on a gauntlet of neural data [14].

In the framework of [17], the objective, learning rule and architecture specify how a task is to be solved by an artificial or biological neural network. Implicit in the framework is a fourth critical ingredient: the dataset, or distribution of training examples. Our work focuses on how an objective, learning rule and dataset interact to form representations similar to the dorsal stream. In contradistinction with previous work, we focus on a single architecture of 3D ResNets, a coarse approximation to early and intermediate visual processing stages, highlighting the formative role of objective, learning rule and dataset in the creation of useful representations for action.

**Maximizing stimuli reveal selectivity**    Systems identification has long been used in systems neuroscience to estimate preferred stimuli in different brain areas [55, 56, 43, 46, 51, 20, 47, 77, 44]. More recently, systems identification has been used to better understand mechanisms of selectivity in deep neural nets [12]. Given the breadth of available systems identification results in brains, we suggest that systems identification is a particularly powerful tool to relate brains and artificial neural nets, especially when combined with benchmarking: it can offer clues as to why certain networks perform better than others. In this article, we identified direction selectivity in the first layer as a strong clue that networks develop good motion representations.

**Action recognition is poorly aligned to the dorsal stream**    Our work shows that ANNs trained on the standard computer vision task of action recognition fail to learn motion representations that correlate with single neurons in MT and MST. [78] reported that on Kinetics400 and UCF101, a single image is sufficient to get within 6% of the action recognition accuracy of a full image sequence, indicating that motion has a limited role in action recognition in these datasets. Motion selectivity can be reintroduced via a parallel optic flow pathway [79] or by enforcing that the network reproduce dense optic flow following early layers [80, 81], with modest improvements in classification accuracy. Our benchmarks strongly suggest that current action recognition datasets can be solved without motion and that good motion representations don't emerge from supervised learning on them alone.

**Self-supervision through cross-modal prediction**    We train DorsalNet in a supervised way. From the agent's perspective, however, corollary discharges of the motor plan are available, as well as vestibular inputs. Thus, the objective can be viewed as a self-supervised objective which aims to predict one modality or channel of the input from the other, in line with other proxy tasks including colorization and audiovisual alignment [82, 83, 84]. Because multisensory integration and corollary discharges are ubiquitous across mobile animals [85], self-supervision through cross-modal prediction could be potentially widely used across species to learn useful representations.

**Evolution and learning in sensory systems**    Thompson [86] identifies four set of constraints against which in silico models of sensory systems can be evaluated:

- Whether it can perform a relevant task
- Whether it accounts for neural activity

- Whether it is biologically plausible
- Whether it could have evolved

We presented a model of the dorsal stream that is trained to estimate self-motion. It accounts for neural responses in 3 different areas, taken from 5 different datasets. The model weights can be learned by the agent through biologically plausible self-supervision, since the approximate parameters of self-motion are known to the agent, via corollary discharges and vestibular and proprioceptive inputs [28]. Self-motion estimation is particularly important for gaze stabilization, which evolved in tandem with the earliest visual functions [87, 88], and continues to be necessary for visual processing, including that performed in the ventral pathway [89]. Given this evolutionary pressure, some aspects of the dorsal pathway are likely hard-coded in the genome, while others are learned through development [90]; further work will focus on better understanding the relative role of evolution vs. learning in dorsal stream processing. This work and its follow-ups thus have the potential to elucidate long-standing questions about how sensory systems evolved.

## Acknowledgments and Disclosure of Funding

This work was supported by a NSERC (Discovery Grant: RGPIN-2020-05105; Discovery Accelerator Supplement: RGPAS-2020-00031), Healthy Brains, Healthy Lives (New Investigator Award: 2b-NISU-8; Innovative Ideas Grant: 1c-II-15), and CIFAR (Canada AI Chair; Learning in Machine and Brains Fellowship). CCP was funded by a CIHR grant (MOP-115178). This work was also funded by the Canada First Research Excellence Fund (CFREF Competition 2, 2015-2016) awarded to the Healthy Brains, Healthy Lives initiative at McGill University, through the Helmholtz International BigBrain Analytics and Learning Laboratory (HIBALL).

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
