# A Appendix

| Parameter | Value |
|---|---|
| Environments | AirSimNH, TrapCamera |
| Dataset size | 39645 movies, 112x112x10 frames |
| Nominal FPS | 30 |
| Heading (yaw) | VonMises$(0, 2.5)$ (rad) |
| Heading (pitch) | VonMises$(0, 16)$ (rad) |
| Head rotation (yaw) | Normal$(\sigma = \pi/6)$ (rad/s) |
| Head rotation (pitch) | Normal$(\sigma = \pi/18)$ (rad/s) |
| Walking speed | Uniform$(0, 3)$ (m/s) |
| Height from ground | Uniform$(1.4, 2)$ (m) |
| Step size | 0.003 |
| Training epochs | 100 |
| Layers | 0: 64 7x7x5 conv filters, stride 1x1x1 |
| | 1: leaky ReLU, 3x3x1 maxpooling, 2x downsampling, batch norm |
| | 2: residual block |
| |    branch 1: 64 filters projected to 32 via 1x1x1 convs |
| |    branch 2: 32 1x1x1 filters, 8 3x3x1, 32 1x1x1, batch norm |
| | 3: residual block, 32 1x1x3 filters, 8 3x3x1, 32 1x1x1, batch norm |
| | 4: residual block, 32 1x1x1 filters, 8 3x3x1, 32 1x1x1, batch norm |
| | 5: residual block, 32 1x1x3 filters, 8 3x3x1, 32 1x1x1, batch norm |
| Boosting step size | 0.1 |
| Boosting max iterations | 100 |

Table S1: Airsim dataset and training parameters

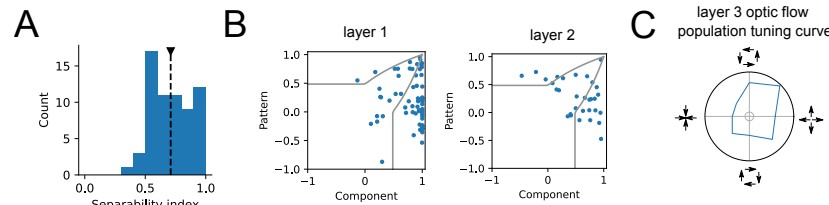

Figure S1: A: Separability index of layer 1. B: pattern index for layers 1 and 2. C: population curves for optic flow in layer 3

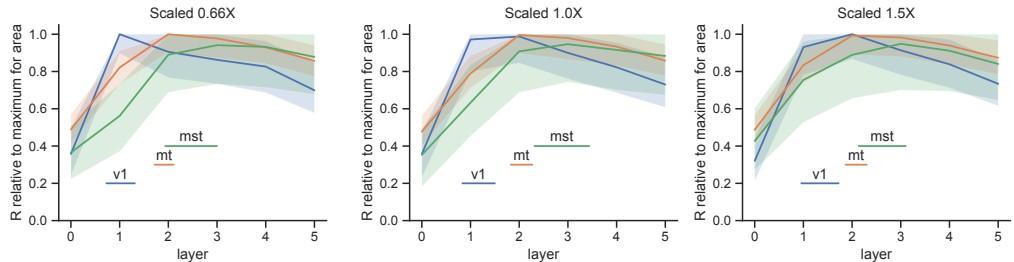

Figure S2: Alignment between layers of DorsalNet and datasets when resizing stimuli. V1 alignment shifts slightly higher as scale is increased, as expected. Alignment is nevertheless broadly similar across different scales.

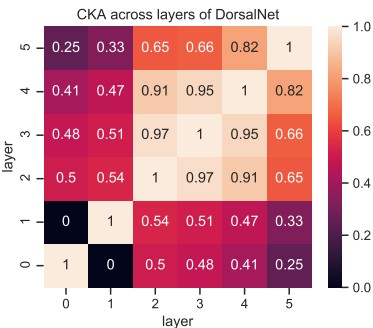

Figure S3: CKA across layers. We used a battery of all stimuli from the airsim dataset to compare representations across layers. We extracted the response of the central pixel of a representation in a given layer and computed alignment between internal representations using centered kernel alignment (CKA) [69]. 0 indicates no alignment, while 1 indicates perfect alignment between layers.

| metric area | overall | pitch | yaw | rotation pitch | rotation yaw | speed |
|---|---|---|---|---|---|---|
| v1 | -0.39 | 0.13 | 0.11 | -0.36 | -0.54 | -0.12 |
| mt | -0.66 | -0.05 | -0.02 | -0.51 | -0.64 | -0.40 |
| mst | -0.53 | 0.05 | 0.05 | -0.51 | -0.69 | -0.13 |

Table S2: Correlation between loss on heading task and performance on data from different areas across models and layers. Performance on predicting head rotation parameters (rotation pitch and rotation yaw) is most correlated with match to different brain areas.

| | | V1 | | MT | | MST |
|---|---|---|---|---|---|---|
| | scaling | pvc1 | pvc4 | mt1 | mt2 | mst |
| motionnet | 0.66X | .303 (.044) | **.373** (.041) | .221 (.018) | .306 (.016) | .403 (.052) |
| | 1X | .276 (.042) | **.364** (.039) | .238 (.018) | .333 (.016) | **.441** (.053) |
| | 1.5X | **.343** (.040) | **.371** (.039) | **.252** (.019) | .346 (.016) | **.452** (.050) |
| dorsalnet | 0.66X | **.358** (.041) | **.380** (.040) | .245 (.018) | **.388** (.016) | **.460** (.056) |
| | 1X | **.364** (.043) | **.370** (.039) | **.251** (.019) | **.381** (.017) | **.454** (.054) |
| | 1.5X | **.389** (.034) | **.359** (.038) | **.252** (.020) | .370 (.017) | .411 (.052) |

Table S3: Relative performance of DorsalNet and MotionNet across different scalings of the input, measured with ridge regression. MotionNet generally benefits from scaling up the videos (1.5X), presumably because of its large second layer receptive fields (27x27). DorsalNet performance is relatively constant across scalings. Table shows normalized pearson correlation (R; see Methods for definition) of different models with different input scaling on different datasets.

| | scaling | MT mt1 | mt2 | MST mst |
|---|---|---|---|---|
| motionnet | 0.66X | - | - | .336 (.050) |
| | 1X | .159 (.012) | .284 (.012) | .361 (.048) |
| | 1.5X | .160 (.011) | .298 (.012) | .385 (.047) |
| dorsalnet | 0.66X | - | - | **.464** (.054) |
| | 1X | **.228** (.017) | **.370** (.016) | **.474** (.051) |
| | 1.5X | **.230** (.017) | .362 (.016) | .434 (.051) |

Table S4: DorsalNet quantitatively performs best across the dorsal stream across different scalings, as measured with boosting after downsampling. Table shows normalized pearson correlation (R; see Methods for definition) of different models with different input scaling on different datasets.

| | scaling | V1 pvc1 | pvc4 | MT mt1 | mt2 | MST mst |
|---|---|---|---|---|---|---|
| motionnet | 0.66X | .371 (.048) | **.319** (.036) | .157 (.012) | .258 (.013) | .345 (.049) |
| | 1X | .426 (.050) | **.311** (.034) | .158 (.012) | .271 (.012) | .359 (.048) |
| | 1.5X | **.460** (.051) | .313 (.036) | .158 (.012) | .282 (.012) | .365 (.047) |
| dorsalnet | 0.66X | .471 (.051) | **.355** (.038) | .212 (.016) | **.353** (.016) | **.435** (.052) |
| | 1X | **.491** (.049) | .313 (.039) | **.217** (.016) | **.348** (.016) | .415 (.055) |
| | 1.5X | **.503** (.051) | **.313** (.034) | .209 (.016) | .328 (.016) | .356 (.052) |

Table S5: DorsalNet quantitatively performs best across the dorsal stream across different scalings, as measured with boosting after subsampling. Table shows normalized pearson correlation (R; see Methods for definition) of different models with different input scaling on different datasets.

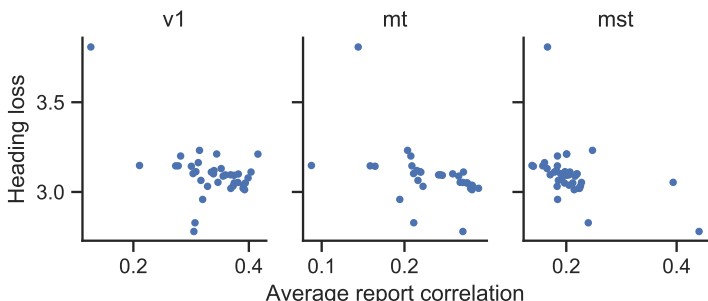

Figure S4: Correlation between heading loss and performance on dorsal stream datasets across networks and layers. Networks and layers which perform better at heading discrimination tend to better match the dorsal stream.

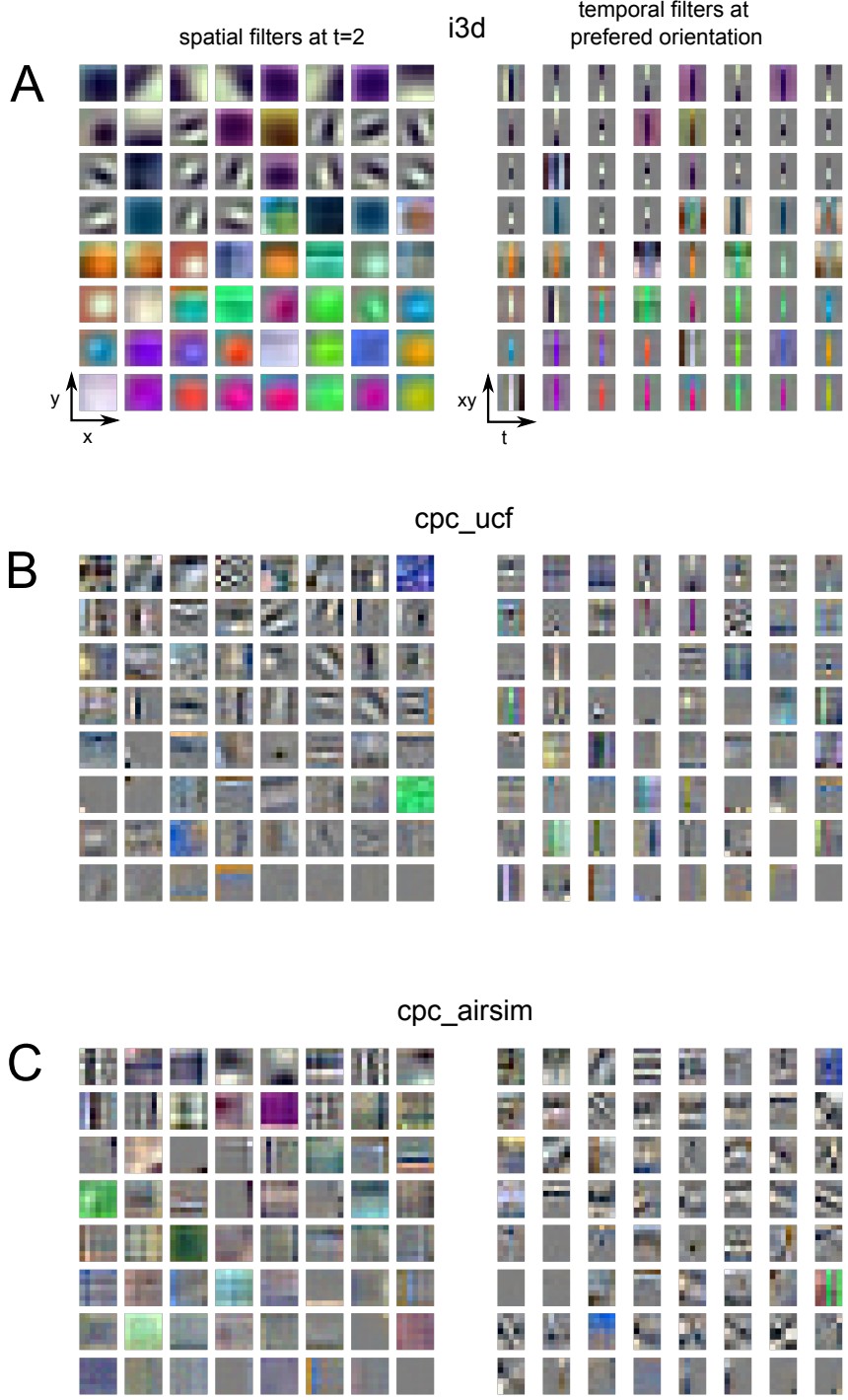

Figure S5: First layer filters for alternative networks. i3d and CPC on UCF learn orientation selectivity but not direction selectivity. CPC on Airsim learns both.