# OpenReview forum: "Your head is there to move you around: Goal-driven models of the primate dorsal pathway"
_NeurIPS.cc/2021/Conference — NeurIPS 2021 Spotlight_

### Official Review · Reviewer_mYCG · 2021-07-11

**Rating:** 8
**Confidence:** 4

**Summary:**

This paper positions itself in an important gap in the computational neuroscience literature which uses artificial neural networks as models of biological vision. The paper uses first-person perspective video stimuli to train a 3d ResNet model to predict self-motion parameters (namely head rotation and velocity) with contrastive predictive coding. The paper examines layer-by-layer the properties of the learned representations and compares them qualitatively and quantitatively to a database of neural recordings from the dorsal stream of non-human primates, recorded while those primates were viewing videos, or classical psychophysical visual stimuli. The authors find that their model learns representations that are more similar to the primates than baseline models or the same model trained on a different task. Finally the authors find that the networks which are functionally achieve  the best self-motion estimation performance are also those that match biological dorsal stream neural data the best.

**Limitations And Societal Impact:**

Yes.

**Main Review:**

This paper is an original and complete contribution to the field which helps to address an important gap in models of biological perception. The approach to train on simulated egocentric video footage and estimate motion is novel for computational neuroscience work, and the layer-by-layer connections between the resulting representations in the ANN and primate NNs were well explored.
Finally the discovery that the ANN models that perform best at predicting self-motion also form the representations most closely aligned with those in the primate dorsal stream is an exciting addition to the literature.

The work was also clearly presented, although as a very minor point more detailed figure legends would be helpful to communicate the main point of each figure more readily to the reader. Note also as an aside a missing 'TODO' link on line 203 which can be easily fixed.

**Time Spent Reviewing:**

2

---

> ### Author Response · Authors · 2021-08-10
> **Reply to reviewer 4**
>
> Thank you for the thoughtful review.
> > more detailed figure legends would be helpful to communicate the main point of each figure more readily to the reader
>
> We agree, and some of the other reviewers made similar comments. We will add more details to the legends in the camera-ready version.
> > Note also as an aside a missing ‘TODO’ link on line 203
>
> Thanks for noticing this, we will fix this in the camera-ready version.

---

### Official Review · Reviewer_X8u7 · 2021-07-12

**Rating:** 7
**Confidence:** 4

**Summary:**

In this paper, the authors develop a new goal-driven model of the primate dorsal pathway based on a ResNet architecture trained to infer self motion parameters. They show that its units show properties reminiscent of properties of neurons found along the dorsal hierarchy. They benchmark this model extensively against other models including object recognition trained ones.

**Limitations And Societal Impact:**

Yes

**Main Review:**

In general, this is a well-written paper, with a clear structure and well-done experiments. A few details could be improved (see below), but the main point of criticism one could voice against this paper is that of novelty and advance over previous work. Given the hype about the ventral stream goal driven models, the present paper seems like a relatively minor extensions. However, I think it is still worthwhile doing, and doing it well (as in the present work). The authors argue for the importance of self-motion cues in the dataset and the task being self motion inference, yet the quantitative advantage over MotionNet (their ref 26) seems small (Table 3, last two lines). Maybe the authors could elaborate on why they still think their task and network are superior in modeling dorsal stream neurons than MotionNets (which seems to use simple displaced images as stimuli and seems to be a bit shallower).

Additional comments:
- Why was a ResNet used as a base model?
- For the alignment between networks and neurons, why was L2 penalized regression used and not L1 regularized?
- How many video segments were generated?
- Line 96: "the" missing in front of Unreal
- The authors call their approach self-supervised in the abstract. Why? To me this seems just a DNN with known ground truth.
- The tuning index in line 170 is likely higher than usual due to the lack of noise
- Why do the correlation values in Fig. 2 go beyond 1? Rather use non-parametric CIs.
- The results in Line 203ff are not obvious from Fig. 2, maybe mark the maximum
- For Table 3, what is the number of experiments and what is the SE over?


**Time Spent Reviewing:**

1

---

> ### Author Response · Authors · 2021-08-10
> **Reply to reviewer 3**
>
> Thank you for the thoughtful review.
> > Maybe the authors could elaborate on why they still think their task and network are superior in modeling dorsal stream neurons than MotionNets (which seems to use simple displaced images as stimuli and seems to be a bit shallower).
>
> As suggested by the reviewer, we performed an additional experiment with a sparse regression, implemented through boosting, to evaluate the alignment of networks and neurons. In that case, we see a larger gap between DorsalNet and MotionNet, which indicates that individual subunits of DorsalNet are better aligned to individual neurons compared to MotionNet.
>
> * MotionNet: mean normalized R = .361 (.048)
> * DorsalNet: mean normalized R = **.474** (.051)
>
> Overall, we think that the biggest contribution of DorsalNet, in addition to its performance, is that it gives a plausible explanation for how the supervised signal might occur. Although the observer does not know the true velocity of objects, they do have good estimates of their self-motion, which they can use for supervision. We will mention this in the next revision.
>
> > Why was a ResNet used as a base model?
>
> We used a ResNet backbone because it’s a similar backbone to state-of-the-art models such as SlowFast and CPC, for a fair comparison. We will note this in the text.
>
> > For the alignment between networks and neurons, why was L2 penalized regression used and not L1 regularized?
>
> L2 regularized alignment is frequently used in the literature (e.g. Güçlü and van Gerven, 2015; Kell et al. 2018). However, sparse alignment has the advantage that it better reflects how individual subunits align to individual cells, as opposed to a mixture of individual subunits. We will report on the control experiment replacing ridge regression with boosted regression highlighted above in the final version.
>
> > How many video segments were generated?
>
> 39645 (Table S1)
>
> > Line 96: “the” missing in front of Unreal
>
> Thank you for noticing; we will fix it.
>
> > The authors call their approach self-supervised in the abstract. Why? To me this seems just a DNN with known ground truth.
>
> We agree with the reviewers that this is confusing. We will remove the statement that the network is self-supervised from the abstract.
>
> > The tuning index in line 170 is likely higher than usual due to the lack of noise
>
> Yes, good point, we will mention it in the text.
>
> > Why do the correlation values in Fig. 2 go beyond 1? Rather use non-parametric CIs.
>
> Thank you for pointing this out, we agree this is unintuitive. We originally normalized the correlation so that its maximum would have a mean of 1.0 across bootstrap samples, hence the upper bound could be higher than 1.0. We regenerated the figure applying the normalization in the inner loop of the bootstrap; in this case, none of the CIs go beyond 1.0. We will swap the figure ([preview here](https://festive-chandrasekhar-bd19f0.netlify.app/alignment.png)) in the final version.
>
> > The results in Line 203ff are not obvious from Fig. 2, maybe mark the maximum
>
> Figure 2 shows the confidence intervals of the estimated maximum alignment at the bottom (horizontal lines), we will mention it in the text so it’s easier to see the connection.
>
> > For Table 3, what is the number of experiments and what is the SE over?
>
> SEM is over the cells (number of cells is in Table 1). We will mention it in the table caption.

---

### Official Review · Reviewer_APpw · 2021-07-13

**Rating:** 6
**Confidence:** 4

**Summary:**

The authors propose to use estimation of self-motion from short video snippets as an objective function to train visual representations resembling those of the dorsal stream of the primate visual system. They show that the first layers of small ResNets trained with this objective develop representations similar to areas V1, MT and MST. According to the authors, alternative objectives such as action classification or contrastive predictive coding or classical baselines like a Gabor filter bank did achieve the same performance on the task of predicting neural responses in the dorsal stream areas – a claim I do not fully agree with; see below. The paper proposes an original idea. It is fairly well executed and well written. I do have some reservations and questions, but I'm overall somewhat supportive of the paper – in particular if the authors can address my concerns.


### Update after discussion period
The authors' response addressed some of my concerns. Unfortunately they promised to report on the control regarding scale but did not post the results, so it's hard to assess this point. I therefore keep my score as borderline accept and hope that the authors will report the results in the final version of the paper.

**Limitations And Societal Impact:**

Yes

**Main Review:**

## Strengths
+ Original idea
+ Proposes an image-computable model of a non-trivial task associated with the dorsal stream
+ Model can be used to create in silico predictions that can be tested experimentally
+ Well written and easy to read

## Weaknesses
- Performance difference to the MotionNet baseline is weak at best
- Task seems too simplistic for a model of the dorsal stream
- Relative scale of receptive fields and input stimuli not considered


## Detailed comments on weaknesses

### Performance difference to MotionNet

The claim that inference of self-motion is a crucial task for modeling the dorsal stream doesn't seem to be fully supported by the data. Neither in MST (the most high-level area considered) nor in one of the two MT datasets does the authors' DorsalNet outperform the much simpler MotionNet that is trained on inferring object motion. Don't these results argue against self-motion being important and rather suggest that the dorsal stream tries to resolve ambiguous local motion by inferring which regions belong to the same objects? I'm not super familiar with MotionNet, so please correct me if I'm interpreting those results wrong.


### Task too simplistic?

While estimating an agent's self-motion is certainly one important objective (and likely one of the dorsal stream), calling DorsalNet a model that explains "the properties of dorsal stream neurons" seems like a bit of an overstatement given that two of the three MT/MST datasets were with random dot kinematograms, where motion is the only cue. In the authors' defense, the MT dataset with natural movies is the only one where their model actually shows an advantage, which could be taken to suggest that they have a point. However, on this dataset CPC on the same stimulus performs almost equally well, suggesting that video sequences that feature optic flow related to self-motion might be just as important as the training objective.

So while I think the authors are on the right track with their approach, the publicly available data does not seem to allow strong conclusions at this point. I would suggest toning down the claims and providing a more nuanced discussion of the possible interpretations rather than claiming strong support of their hypothesis, which I don't see.


### Relative scale of receptive fields and input stimuli

Since the models are image-computable and different datasets from different brain areas and labs are combined, I was a bit surprised to not see any discussion of scale. How did the authors decide how large one pixel in the models should be in terms of degrees of visual angle? Neurons have different receptive field sizes depending on eccentricity and brain area. Depending on which input resolution one picks, the models probably perform better or worse, and the conclusion about which layer (or model) predicts neural activity best can change (see, e.g., the work by Cadena et al., PLoS Comp Bio 2019). Unless the authors considered this issue and I missed it, I think it is important to clearly describe the reasoning here and potentially perform some controls to ensure that the inputs were provided at the optimal resolution or the conclusions don't change as a function of input resolution.



## Minor comments

- I wouldn't call the training paradigm "self-supervised". It's using ground truth information from the rendering engine as labels. Self-supervised techniques usually derive their labels from the images themselves by augmentation, cropping other pretext tasks.

- The discussion talks about the "curriculum," but means the dataset. Curriculum is usually rather used to refer to a sequence of training examples that vary over time w.r.t some feature (e.g. complexity), so I was a bit surprised at first until I realized how it's meant.

- It's not clear what the bold numbers in Table 3 mean. Given the SEM values in parentheses it doesn't seem to signify statistical significance. It would be great to clarify.



**Time Spent Reviewing:**

3

---

> ### Author Response · Authors · 2021-08-10
> **Reply to reviewer 2**
>
> Thank you for the thoughtful review.
> > Performance difference to MotionNet
>
> Prior to this study, it would have been difficult to foresee that MotionNet would outperform much larger models trained on computer vision tasks - MotionNet had not been tested on the neural datasets we use here. We think that this observation is itself a valuable contribution to the literature, which we will mention in the revised document.
> > Neither in MST (the most high-level area considered) nor in one of the two MT datasets does the authors' DorsalNet outperform the much simpler MotionNet that is trained on inferring object motion.
>
> Our presentation of the mt1 results obscured the better performance of DorsalNet compared to MotionNet on this dataset - some recordings are much shorter than others, and some cells are less active than others, which causes a lot of shared variance in the prediction accuracy. DorsalNet prediction outperformed MotionNet in 62/84 cells (p < 1e-4, sign test; p < 1e-5, paired z-test).
>
> Part of the challenge with the MST dataset is that the recordings are fairly short, and ridge regression requires a good amount of data to be highly discriminative. We’ve done an additional experiment using sparse regression, implemented through boosting, to align representations rather than ridge regression. In this case, we see a significant gap (p < 1e-5, sign test; p < 1e-8, paired z-test) between DorsalNet and MotionNet in MST:
>
> * MotionNet: mean normalized R = .361 (.048)
> * DorsalNet: mean normalized R = **.474** (.051)
>
> This indicates that individual subunits in DorsalNet are better aligned to individual cells in MST compared to MotionNet. We will add this analysis to the camera-ready version.
>
> > Don't these results argue against self-motion being important and rather suggest that the dorsal stream tries to resolve ambiguous local motion by inferring which regions belong to the same objects?
>
> Overall, we think that the biggest contribution of DorsalNet, in addition to its performance, is that it gives a plausible explanation for how the supervised signal might occur. Although the observer does not know the true velocity of objects, they do have good estimates of their self-motion, which they can use for supervision. We will mention this in the next revision.
>
> > Task too simplistic?
>
> We agree with the reviewer that the datasets - especially the random-dot kinematograms - have significant limitations, which we attempted to address by pooling from all the available data. We will point out that the random dot kinematograms are not very discriminative in the Data Limitations section (lines 264-270). Prior to this study, it would have been hard to justify gathering more data from multiple areas to make the ideal benchmark - we believe this study will motivate experimentalists to gather that benchmark data.
>
> > I would suggest toning down the claims and providing a more nuanced discussion of the possible interpretations rather than claiming strong support of their hypothesis, which I don’t see.
>
> We will tone down the claims in the abstract - in particular line 14, that a contrastive objective could not explain responses - and mention multiple interpretations in the Limitations section.
>
> > Since the models are image-computable and different datasets from different brain areas and labs are combined, I was a bit surprised to not see any discussion of scale. How did the authors decide how large one pixel in the models should be in terms of degrees of visual angle?
>
> Since many of the datasets did not note the center of the receptive field and position of the stimulus, precluding good estimates of scale factor, we simply scaled stimuli to a standard size (112x112). Running a control experiment as suggested by Cadena et al. (2019) is a great idea. We will run it this week and post the results once obtained.
>
> > I wouldn’t call the training paradigm “self-supervised”. It’s using ground truth information from the rendering engine as labels. Self-supervised techniques usually derive their labels from the images themselves by augmentation, cropping other pretext tasks.
>
> We agree with the reviewers that this is confusing. We will remove the statement that the network is self-supervised from the abstract.
> > The discussion talks about the “curriculum,” but means the dataset. Curriculum is usually rather used to refer to a sequence of training examples that vary over time w.r.t some feature (e.g. complexity), so I was a bit surprised at first until I realized how it’s meant.
>
> The reviewer is correct, we were using a non-standard terminology. Thank you for noting this. We’ll use ‘dataset’ instead of ‘curriculum’.
> > It’s not clear what the bold numbers in Table 3 mean. Given the SEM values in parentheses it doesn’t seem to signify statistical significance. It would be great to clarify.
>
> Thank you for pointing this out. We will explain in the table caption why the numbers are in bold: bold indicates that a model is not significantly different from the best performing model according to a paired z-test (p > .05, two-tailed). Much of the variance in the prediction accuracy is due to limitations in the data - some recordings are much shorter than others, and some cells are less active than others. Because of this, a paired z-test is more powerful to discriminate among alternative models than an unpaired z-test. The SEM value - implicitly unpaired - obscures this.

---

> > ### Comment · Reviewer_APpw · 2021-09-01
> > **Was waiting for the promised results**
> >
> > Thank you for the detailed response. It clarifies some of my questions/issues.
> >
> > I was waiting for the results on the scale question, which is why I didn't respond immediately. It would have been great to see the numbers, since it may have strengthened or weakened the claims of the paper.
> >
> > Assuming that adjusting scale doesn't completely change the results (which I hope the authors still do before the paper goes to press), I think the paper is very worthwhile publishing.

---

> > > ### Author Response · Authors · 2021-09-03
> > > **Scaling results**
> > >
> > > Thank you for your patience as we refit the models at multiple scales. We find that changing the scale from .66X to 1X to 1.5X broadly maintains the layer assignments in dorsalnet, with some minor shifts in the expected direction: the median mean layer assignment for V1 is 1.0 at scale .66X, 1.14 at scale 1X, 1.32 at scale 1.5X.
> > >
> > > [Relative alignments figure](https://61323364fece253795f7874d--festive-chandrasekhar-bd19f0.netlify.app/layer-alignment.png)
> > >
> > > We find that performance in dorsalnet is largely optimal at the current scale (1.0X), but motionnet performs better at a larger scale (1.5X), presumably because of its large second layer kernels (27 x 27).
> > >
> > > [Scaling results, ridge](https://61323364fece253795f7874d--festive-chandrasekhar-bd19f0.netlify.app/scaling-regression.png)
> > >
> > > However, in a control analysis in MST, we find that while this closes the gap between motionnet and dorsalnet as estimated with ridge regression, it does not close the gap in performance as estimated by boosting. Thus, there is a better one-to-one correspondence between single units in dorsalnet to single cells in MST.
> > >
> > > [Scaling results, boosting](https://61323364fece253795f7874d--festive-chandrasekhar-bd19f0.netlify.app/scaling-boosting.png)
> > >
> > > We are currently running the same boosting alignment analysis in MT at the 1.5X scale and will update you with the results.

---

### Official Review · Reviewer_VDAo · 2021-07-17

**Rating:** 7
**Confidence:** 3

**Summary:**

This paper evaluated the ability of artificial neural networks with different learning objectives to explain single-neuron responses in the primate dorsal visual pathway. A 3D ResNet trained to predict action motion parameters in simulation videos matched dorsal stream responses better than networks trained for action recognition, suggesting that estimating self-motion may be a main role of neurons in the dorsal stream.

**Limitations And Societal Impact:**

The authors have discussed the main limitations of the work.

**Main Review:**

The authors evaluated a range of neural networks, including one trained to estimate self-motion parameters (DorsalNet), in terms of their match to primate neural data. DorsalNet matched dorsal stream data better than networks trained for action recognition or latent state prediction. The approach of comparing networks with different learning objectives is powerful, and different training sets are also represented across the neural networks tested.

The convergence of results across experiments and stimuli is another strength of the paper. However, the fact that recordings from different brain areas were collected in different datasets is also a challenge when directly comparing results across areas, which could be mentioned more clearly in the paper. This seems particularly important given the small difference between the DorsalNet and MotionNet performance in MST (a single dataset available).

The intermediate representations in layers 1-3 of DorsalNet are interesting in terms of their match to the hypothesized features that drive dorsal stream responses. Figure 2 suggests that the DorsalNet match to dorsal neural data peaks at ~layer 3, and the paper focuses on layers 1-3. I wonder how representations in later layers compare to the intermediate ones, since the decrease in performance in later layers is so small, especially in MST.

Discussion: Although the results do suggest a separation between action recognition and motion estimation, the paragraph in lines 295-305 seems to discuss conclusions that don’t necessarily follow from the data at hand. Since the paper did not look at action recognition with/without motion in the ventral stream, the conclusions here could be rephrased to be more focused.

Finally, it is somewhat unclear in what sense dorsalnet is self-supervised. Although the discussion explains that the self-supervision refers to the goal from the agent’s perspective, the abstract describes the network as having been “trained in a self-supervised manner” – perhaps this could be rephrased to be more specific/accurate.

However, the paper addresses the question of goals in the dorsal stream in an interesting way and paves the way for more direct confirmatory experiments using DNN-generated stimuli/simulation videos.

------------------------------------------------
After author response:

I thank the authors for their thorough response. Interesting to see the results using sparse regression, as well as the layer representation results. I look forward to reading the updated paper!



**Time Spent Reviewing:**

5

---

> ### Author Response · Authors · 2021-08-10
> **Response to reviewer 1**
>
> Thank you for the thoughtful review.
> > However, the fact that recordings from different brain areas were collected in different datasets is also a challenge when directly comparing results across areas, which could be mentioned more clearly in the paper. This seems particularly important given the small difference between the DorsalNet and MotionNet performance in MST (a single dataset available).
>
> Using different stimuli allows us to compare models within a given area, while to some extent making the model comparison more robust to specific choices of stimuli. It does make direct comparisons between areas more difficult. We will mention some of the limitations in these datasets more clearly (lines 264 to 270), namely caveat that absolute comparisons are difficult, and that the MST dataset is not very discriminative across models. Ideally, we’d have a standardized battery of stimuli across all areas, and we hope this manuscript will motivative experimentalists to perform such experiments.
>
> Part of the challenge with the MST dataset is that the recordings are fairly short, and ridge regression requires a good amount of data to be highly discriminative. We’ve done an additional experiment using sparse regression to align representations rather than ridge regression - in this case we do see a significant gap between DorsalNet and MotionNet in MST.
>
> * MotionNet: mean normalized R = .361 (.048)
> * DorsalNet: mean normalized R = **.474** (.051)
>
> This indicates that individual subunits in DorsalNet are better aligned to individual cells in MST compared to MotionNet.
>
> > I wonder how representations in later layers compare to the intermediate ones, since the decrease in performance in later layers is so small, especially in MST.
>
> Thank you for the suggestion. We performed an additional analysis to measure how representations evolve in the network ([figure here](https://festive-chandrasekhar-bd19f0.netlify.app/cka.png)). Using a battery of stimuli from the airsim dataset, we computed alignment between internal representations using centered kernel alignment (CKA; Kornblith et al. 2019). 0 indicates no alignment, while 1 indicates perfect alignment between layers. While layers 0, 1, and 2 are highly distinct, subsequent layer representations are much less so. We will discuss this and include the figure in the supplementary.
>
> > Although the results do suggest a separation between action recognition and motion estimation, the paragraph in lines 295-305 seems to discuss conclusions that don’t necessarily follow from the data at hand.
>
> We agree this section is a little unfocused. We will change the heading to “Action recognition is poorly aligned to the dorsal stream”, and remove the last sentence.
>
> > Finally, it is somewhat unclear in what sense dorsalnet is self-supervised.
>
> We agree with the reviewers that this is confusing. We will remove the statement that the network is self-supervised from the abstract.

---

### Decision · Program_Chairs · 2021-09-27

**Decision:**

Accept (Spotlight)

**Comment:**

This paper received 4 accepts (including a marginal accept). The paper makes an original and complete contribution to the field and addresses an important gap in biological vision. I would add that unlike much of the current work in computational models of vision where the goal appears to simply quantitatively fit model responses to neural data, this paper performs extensive in-sillico electrophysiology on the model responses to highlight qualitative properties using classic stimuli that have been used to characterize the dorsal stream. The AC recommends accepting as a spotlight.